# Cardiac ryanodine receptor distribution is dynamic and changed by auxiliary proteins and post-translational modification

Parisa Asghari[1], David RL Scriven[1], Myles Ng[1], Pankaj Panwar[2], Keng C Chou[3], Filip van Petegem[2], Edwin DW Moore[1]*

[1]Department of Cellular and Physiological Sciences, University of British Columbia, Vancouver, Canada; [2]Department of Biochemistry and Molecular Biology, University of British Columbia, Vancouver, Canada; [3]Department of Chemistry, University of British Columbia, Vancouver, Canada

**Abstract** The effects of the immunophilins, FKBP12 and FKBP12.6, and phosphorylation on type II ryanodine receptor (RyR2) arrangement and function were examined using correlation microscopy (line scan confocal imaging of $Ca^{2+}$ sparks and dual-tilt electron tomography) and dSTORM imaging of permeabilized Wistar rat ventricular myocytes. Saturating concentrations (10 µmol/L) of either FKBP12 or 12.6 significantly reduced the frequency, spread, amplitude and $Ca^{2+}$ spark mass relative to control, while the tomograms revealed both proteins shifted the tetramers into a largely side-by-side configuration. Phosphorylation of immunophilin-saturated RyR2 resulted in structural and functional changes largely comparable to phosphorylation alone. dSTORM images of myocyte surfaces demonstrated that both FKBP12 and 12.6 significantly reduced RyR2 cluster sizes, while phosphorylation, even of immunophilin-saturated RyR2, increased them. We conclude that both RyR2 cluster size and the arrangement of tetramers within clusters is dynamic and respond to changes in the cellular environment. Further, these changes affect $Ca^{2+}$ spark formation.

**\*For correspondence:**
edwin.moore@ubc.ca

**Competing interests:** The authors declare that no competing interests exist.

## Introduction

Ryanodine receptors are large (~2.2 MDa) homotetrameric $Ca^{2+}$-activated $Ca^{2+}$ ion channels located in the endoplasmic and sarcoplasmic reticulum of virtually every cell (*Lanner et al., 2010*; *Van Petegem, 2015*; *Meissner, 2017*). They are distributed in discrete clusters whose various shapes and sizes contribute to the spatial and temporal characteristics of $Ca^{2+}$ release. Of the three highly homologous isoforms, ventricular myocytes only express the type II ryanodine receptor (RyR2) and its intracellular location is restricted to the junctional sarcoplasmic reticulum (jSR) membrane in close apposition to the sarcolemma (*Carl et al., 1995*; *Sun et al., 1995*). The small distance between the two membranes,~15 nm, provides a space of restricted diffusion where a small influx of $Ca^{2+}$ through sarcolemmal L-type $Ca^{2+}$ channels ($Ca_v1.2$) can open the RyR2 via $Ca^{2+}$-induced $Ca^{2+}$ release (CICR) (*Franzini-Armstrong, 2018*). This architecture is critical for normal cardiac function (*Louch et al., 2006*; *Song et al., 2006*).

In diastole, and in the absence of $Ca^{2+}$ influx through $Ca_v1.2$, RyR2 open stochastically but with a very low probability (*Satoh et al., 1997*). A single tetramer can either open without further effect, or it can activate other RyR2 through inter-RyR2 CICR producing a $Ca^{2+}$ spark (*Cheng et al., 1993*; *Santiago et al., 2010*; *Sobie et al., 2006*). This diastolic SR $Ca^{2+}$ release, with and without sparks, is normal and asymptomatic in a healthy individual, but it is elevated in both acquired and inherited

cardiac diseases such as hypertrophy, heart failure (*Song et al., 2005*) and catecholaminergic polymorphic ventricular tachycardia (CPVT) (*Uchinoumi et al., 2010*), leading to increased morbidity and mortality. Less $Ca^{2+}$ in the SR results in a negative inotropic effect, and the elevated release reduces the rate of decline of the $Ca^{2+}$ transient producing a negative lusitropic effect. Removing the excess $Ca^{2+}$ from the myoplasm is energetically costly since it utilizes ATP. If the leak is sufficiently large, the $Na^+/Ca^{2+}$ exchanger can depolarize the cell and seed a potentially lethal arrhythmia, a common cause of death in cardiac disease (*Bers, 2014*). For these reasons, elevated SR $Ca^{2+}$ release is a significant driver behind current efforts to understand how RyR2 are regulated and how they function.

Abnormal RyR2 activity and $Ca^{2+}$ release has also been linked to many other disorders, including atrial fibrillation (*Dobrev et al., 2011*), cognitive dysfunction (*Liu et al., 2012*), Alzheimer's disease (*Stutzmann et al., 2006*), and diabetes mellitus (*Santulli et al., 2015*). All of these, in addition to heart failure and arrhythmia, have been suggested to originate, at least in part, through RyR2 phosphorylation and a decrease in its affinity for the immunophilin FKBP12.6 (calstabin 2) (*Kushnir et al., 2018*; *Marx et al., 2000*). While this mechanism remains contentious (*Camors and Valdivia, 2014*; *Xiao et al., 2004*), as do the actions of the FKBPs even in the absence of disease (*Gonano and Jones, 2017*), the adverse effects of pathologic $Ca^{2+}$ release through abnormally functioning RyR2 has been well established.

Recently, pathologically induced changes in the RyR2 cluster sizes was identified as the likely cause of cerebral artery smooth muscle tone dysfunction in the mdx mouse (*Pritchard et al., 2018*) and in rat ventricle the probability of a $Ca^{2+}$ spark was found to be correlated to the size of the cluster (*Galice et al., 2018*). These results suggest that tetramer movement and cluster size might be variables that contribute to regulating RyR2 function in both health and disease. Intuitively, the receptors' relative positions should also be of importance in regulating the positive feedback of inter-RyR2 CICR and could therefore be modulated, but mathematical models have produced conflicting results (*Cannell et al., 2013*; *Tanskanen et al., 2007*; *Iaparov et al., 2019*). Empirical studies are few, but phosphorylation and $Mg^{2+}$ were seen to reorient the tetramers and their relative positions were correlated with the $Ca^{2+}$ spark frequency (*Asghari et al., 2014*). We therefore set out to test the impact of FKBP binding, hypothesizing that it would alter distribution and spark frequency, and to examine the effect on RyR2 cluster sizes.

We used correlative microscopy to examine the position and orientation of RyR2 tetramers relative to each other, and $Ca^{2+}$ sparks from within the same cells, and found that exposing the tetramers to a saturating concentration of either FKBP12 or FKBP12.6 resulted in a reduced SR $Ca^{2+}$ release and a more densely packed side-by-side tetramer array. Exposing similarly treated myocytes to cell-wide phosphorylation produced the opposite result; increased SR $Ca^{2+}$ release and more widely spaced tetramers. dSTORM microscopy observed widespread changes in RyR2 cluster sizes and in the number of tetramers per cluster that supported and extended those obtained from correlative microscopy.

We conclude that RyR2 are not distributed randomly or independently of each other on the jSR membrane. Rearranging the tetramer array and changing the cluster sizes is part of the normal operation of the mammalian ventricle and are likely mechanisms that alter the probability of $Ca^{2+}$ spark formation.

## Results

RyRs can have from zero to 4 FKBPs associated with them. The amount and the proportion of each isoform in the native cell is uncertain, and the level of saturation of the binding sites is estimated to be between 45–100% (*Guo et al., 2010*; *Jeyakumar et al., 2001*; *Zissimopoulos et al., 2012*). To isolate the effects of each isoform, and to ensure saturation of the RyR2 tetramers, we delivered 10 µmol/L of purified FKBP12 or FKBP12.6 to the permeabilized myocytes. The results thus obtained illuminate the inherent ability to cluster and move for those RyR2 channels with four FKBPs bound.

The images in *Figure 1A* are representative line scans of $Ca^{2+}$ sparks recorded from saponin permeabilized myocytes at room temperature in a solution containing 100 nmol/L $Ca^{2+}$ and 1 mmol/L $Mg^{2+}$, conditions comparable to those in a resting myocyte. The images have been placed into an array with columns designating the treatment (buffer [Control]; 10 µmol/L FKBP12; 10 µmol/L FKBP12.6), and rows indicating whether the cells were (bottom) or were not (top) exposed to the phosphorylation cocktail.

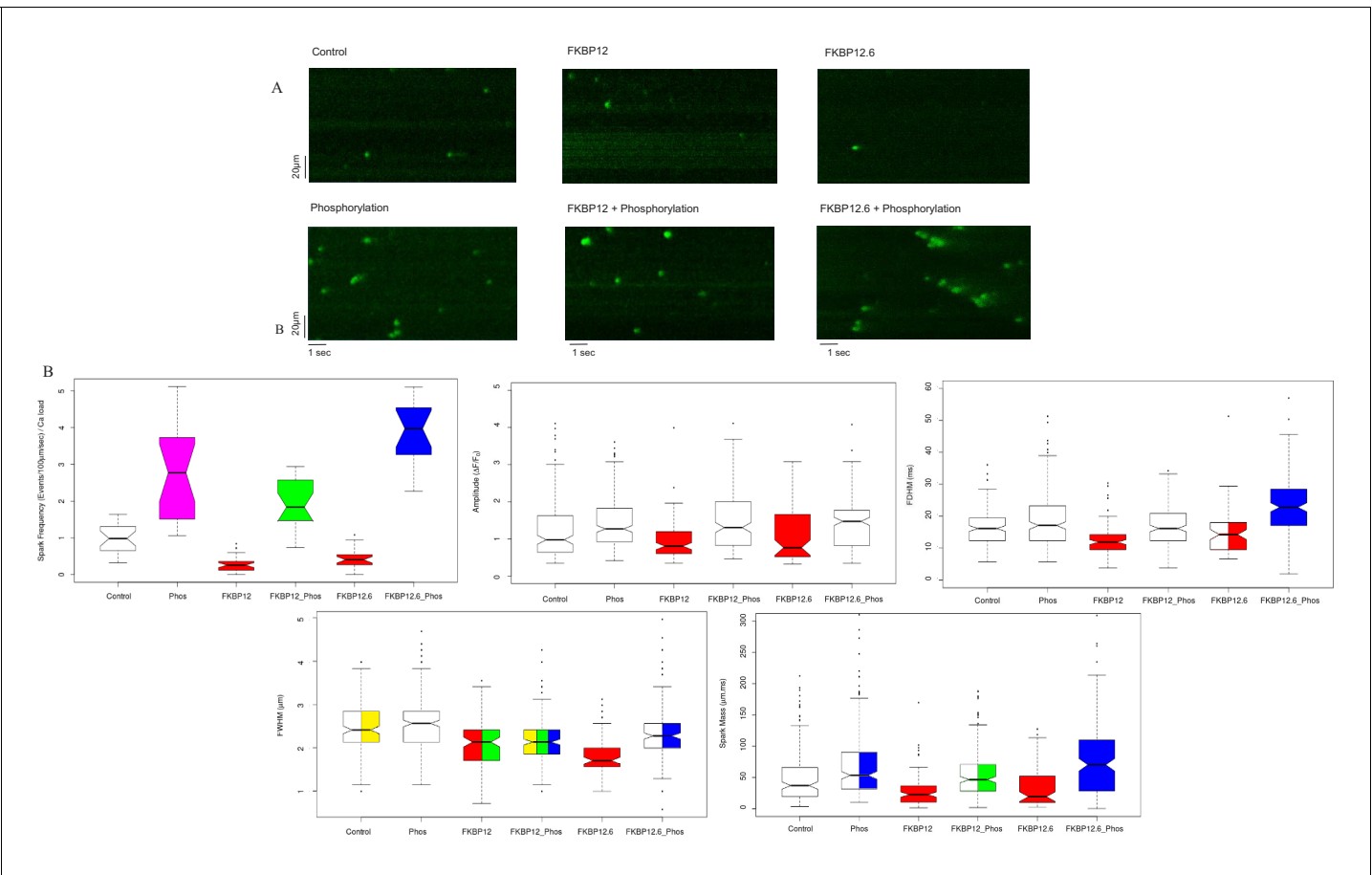

**Figure 1.** Calcium Sparks. (**A**) Representative line-scan images of Ca$^{2+}$ sparks recorded from permeabilized rat ventricular cardiomyocytes under the indicated conditions. (**B**) Box and whisker plots of the aggregated Ca$^{2+}$ spark data under the indicated experimental condition (i) Frequency, normalized to SR content. (ii) Amplitude, (iii) FDHM (ms) (iv) FWHM (μm) (v) Spark mass (amplitude x FDHM x FWHM). The bottom and top of the boxes are the first and third quartiles of the data, and the central line is the median. The data are presented with a color code: those treatments that were significantly different from each other (p<0.05) have different colors, while those that were not have the same color. Quartiles of the box and whisker plots as well as a full statistical analysis is available in *Figure 1—source data 1*. The number of rats and cells used, and the number of sparks analysed, for each of the experimental groups is in *Figure 1—source data 2*.

The online version of this article includes the following source data for figure 1:

**Source data 1.** Statistical analysis of calcium spark data.
**Source data 2.** Number of sparks, cells and rats examined.

The distributions of most of the Ca$^{2+}$ spark parameters were not normal, so they were displayed as box plots (*Figure 1B*). For each spark parameter in the six experimental groups there are fifteen different comparisons, and for this reason we used a color code to indicate statistical differences. Data that were not significantly different from each other share the same color, while those that are significantly different, do not. Some box plots contain two or more colors; for example, the spark mass plot in *Figure 1B*, phosphorylation is bi-colored because it was not significantly different from control (white) or FKBP12.6 plus phosphorylation (blue), but the latter two were significantly different from each other. The probabilities for the comparisons for each parameter and whether they have statistical significance is shown in *Figure 1—source data 1*; an explanation of the table columns is given in the Materials and methods section.

The immunophilins significantly reduced the spark frequency and most of the Ca$^{2+}$ spark parameters relative to control. Comparing control, FKBP12 and FKBP12.6 treated cells to their phosphorylated counterparts demonstrated that phosphorylation significantly increased many of the Ca$^{2+}$ spark parameters, although the degree depended on the particular immunophilin. FKBP12 plus

phosphorylation produced far fewer sparks than FKBP12.6 plus phosphorylation (p<10$^{-12}$), possibly reflecting their differing effects on RyR2 gating (*Gonano and Jones, 2017*). The spark frequency of phosphorylation alone lies between the two and is significantly different from both (p<2×10$^{-4}$). The number of sparks analyzed, cells examined, and animals used are in *Figure 1—source data 2*.

Given these results, and those from our previous study, we hypothesized that changes in the tetramers' distribution would accompany the functional changes and used correlation microscopy to examine dyads in the same cells from which we had recorded the Ca$^{2+}$ sparks.

RyR2 is a homotetramer resembling a square-capped mushroom about 27 nm wide and 12 nm deep (*Peng et al., 2016*). Its size, shape and position on the jSR membrane makes it one of the few proteins that can be identified in an electron micrograph without the aid of additional markers (*Franzini-Armstrong, 1994*), a significant advantage of this technique. We examined the distribution of RyR2 tetramers in junctions using the Multi Planner Viewer (Amira software), simultaneously viewing three orthogonal sections of a selected junction enabling a 360-degree view of the dual aligned 3D data sets, and of every RyR2 (*Figure 2*). The 3D view is essential because the dyadic cleft has a complex geometry. It is usually uneven and often curved such that a single plane cannot view all of the tetramers at once. Multiple planes and multiple points of view are required.

Using Amira, a single plane from a dual-tilt tomogram is displayed in *Figure 2A* (control) showing the jSR (single arrow) adjacent to a t-tubule (double arrow). Viewpoints within the tomogram's volume are determined by orthogonal planes: XY; red (*Figure 2A,C*), YZ; green (*Figure 2B*) and XZ; blue (*Figure 2D*). A single XZ plane that bisects the cleft enabling all of the tetramers' positions and orientations to be visualized and mapped is usually impossible because of the uneven and undulating surface of the jSR membrane. We therefore positioned a XZ plane as parallel to the jSR surface as possible and, progressing in 0.5 nm increments, moved through the cleft to identify a plane where a given tetramer was clearly seen; the tomogram was then tilted in X and Y until its four corners were clear, at which point a red box of the appropriate size (27 nm on a side) was manually positioned over the tetramer and the image was captured. This was done for each tetramer in the tomogram. These images were then used to inform the positioning of the boxes in the RyR_fit program. For displaying all of the RyR2 in the tomogram, we used RyR_fit to manually insert all of the red boxes onto a single XZ plane with a tilt of 0°. Not all of the tetramers are clearly visible in a single image plane, so some of the boxes are atop what appear to be oddly shaped blobs. For clarity, we redisplay the boxes on a black background. Of the 28 XZ planes that spanned this cleft (*Figure 2D*), the selected tetramer (arrows) was clearly seen in planes 2–6, and plane 4 (outlined in blue) was tilted in X and Y to visualize its corners (*Figure 2E*); in this instance rotations of ~5° in X and 0° in Y gave a clear result (*Figure 2Fi*). All of the red boxes, identifying the tetramers' positions and orientations, were displayed on a single XZ plane (*Figure 2Gi*) and then on a black background (*Figure 2Gii*); the tetramer fitted in *Figure 2D and E* is highlighted with an asterisk. Mapping the position and orientation of the tetramers labelled a, b and c (*Figure 2Fii*) is shown in *Figure 2—figure supplement 1*. A 3D model of this tomogram's RyR2, and its jSR and t-tubule membranes, can be seen in the accompanying movie, *Video 1*.

Measurement of the tetramers' centre-to-centre nearest neighbor distances (NND), also derived from RyR_fit, is displayed in *Figure 2Hi*, and a histogram of all the results obtained from the control group (five tomograms, 98 tetramers) is in *Figure 2Hii*. The histogram has a bimodal distribution with modes at 28 and 34 nm, comparable to our previous results (*Asghari et al., 2014*). As can be seen in *Figure 2Hi*, the first mode corresponds to tetramers that are side-by-side, while the second shows tetramers that are mostly in a checkerboard-like arrangement. Using our classification criteria, 50.0% of the tetramers had neighbors that were only in a checkerboard configuration and 27.6% had neighbors that were only side-by-side. 8.2% had neighbors in both side-by-side and checkerboard configurations while 14.3% were isolated (*Table 1*).

Tetramer arrangements recorded in each of the indicated experimental conditions using the RyR_fit program which is available in *Table 1*, *Source code 1*.

Comparable images of a myocyte treated with the phosphorylation cocktail are shown in *Figure 3* and a model of the complete tomogram is in the movie *Video 2*. The images in *Figure 3* demonstrate that the tetramers adopted a more ordered and less densely packed configuration when phosphorylated, with 75.0% of them in a checkerboard and only 6.3% that were side-by-side (64 tetramers, four tomograms; *Table 1*). These changes resulted in a unimodal histogram with a single broad peak centred at 34 nm.

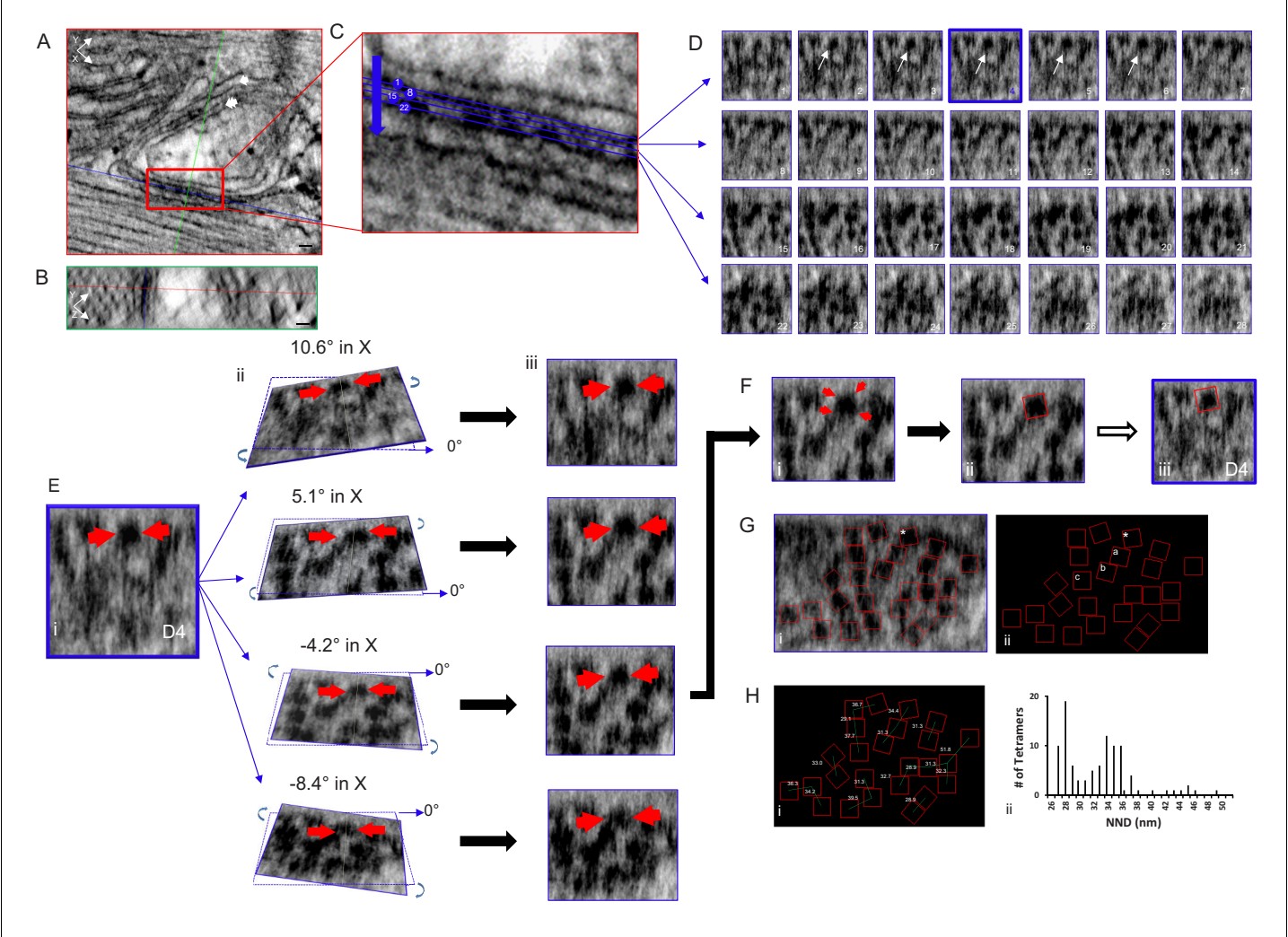

**Figure 2.** Tomogram of a Rat Ventricular Myocyte Dyad. Orthogonal views, (**A**) XY and (**B**) YZ, 1nm-thick, through a dyad, with the intersecting planes (red line – XY plane, blue line - XZ plane, green line - YZ plane) positioned over a single RyR2. Single arrow – jSR, double arrow – t-tubule. Scale bars, 30 nm. (**C**) Magnified view of the boxed area in (**A**), 4 of the 28 XZ planes from which images were acquired are highlighted. (**D**) Sections of all 28 image (XZ planes) acquired through the width of this dyadic cleft. A single tetramer was identified for mapping (arrows, planes 2–6). Plane D4 (blue) was selected. (**E**) (i) Magnification of image plane D4; arrows indicate the selected tetramer. ii Rotations about the X axis, at the indicated angles, through the middle of the tetramer. iii XZ views of the images in ii. (**F**) (i) The tetramer's four corners were visible with a 5.1˚ rotation in X and 0˚ in Y. (ii) A red box 27 nm on a side was placed over the tetramer, then transferred to the original image (iii). (**G**) Image of a single XZ plane of the tomogram (i) and a black background (ii) onto which the position and orientation of all of the tetramers has been superimposed. *the tetramer identified in (E and F). Mapping the position and orientation of the tetramers labelled a, b and c is in *Figure 2—figure supplement 1*. (**H**) (i) NND between the tetramers. (ii) Histogram of the NND of all of the tetramers examined in the control group (98 tetramers, five tomograms, *Table 1*). C++ code for RyR_fit program is available in *Source code 1*.

The online version of this article includes the following figure supplement(s) for figure 2:

**Figure supplement 1.** Localization and Distribution of 3 RyR2 Tetramers Identified in *Figure 2 (a,b and c)* on the Surface of the jSR.

Permeabilized myocytes treated with saturating concentrations of either FKBP12 (*Figure 4A*) or FKBP12.6 (*Figure 4C*) produced dramatic effects on the tetramer distribution, orienting them into a more densely packed side-by-side configuration; 57.1% for FKBP12 (119 tetramers, five tomograms) and 82.8% for FKBP12.6 (122 tetramers examined; *Table 1*). Both proteins shifted the histograms of the centre-to-centre NND to a single mode with a peak at 28 nm (*Figure 4Aiii* and 4Ciii), reflecting the preponderance of the side-by-side configuration when the tetramers were completely saturated

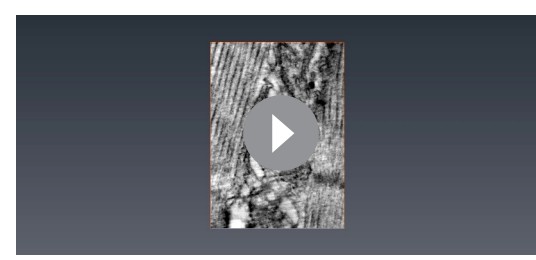

**Video 1.** Dyad of a control myocyte:tomogram and model.
https://elifesciences.org/articles/51602#video1

with either of these proteins. Models of these complete tomograms are in the movies *Video 3* (FKBP12) and *Video 4* (FKBP12.6).

Exposing immunophilin-treated tetramers to cell-wide phosphorylation partially reversed the proteins' effects (*Figure 4B and D*), but seemed more effective in separating tetramers treated with FKBP12.6 than with FKBP12. As summarized in *Table 1*, FKBP12.6 plus phosphorylation increased the percentage of tetramers in a checkerboard to 71.3%, with side-by-side tetramers reduced to only 5.2% (115 tetramers examined in eight tomograms). The comparable values for FKBP12 plus phosphorylation were 60.5% checkerboard and 13.2% side-by-side (129 tetramers examined in seven tomograms). Models of these tomograms are in *Video 5* (FKBP12 + phosphorylation cocktail) and *Video 6* (FKBP12.6 + phosphorylation).

To determine which, if any, of the tetramer distributions could have been derived from the same population, the histograms were expressed as cumulative distribution functions (CDF; *Figure 5*) to enable statistical analysis using the k-sample Anderson-Darling test. In all cases, phosphorylation produced a highly significant rightward shift in the distribution, showing that phosphorylation overwhelms the effect of the immunophilins. In particular, FKBP12.6 shifted from being the leftmost curve to being the rightmost. For FKBP12, the shift was less dramatic and it was noticeable that the effects on spark frequency were significantly less than those for FKBP12.6 (*Figure 1B*).

Tomograms currently provide the highest possible resolution with which to view the tetramers, but to determine their position and orientation the images must be limited to a small area within a single dyadic cleft. To determine whether the results were truly representative of wide-spread changes in the cells we imaged the distribution of RyR2 near the myocyte surface using our custom-built superresolution microscope in near-TIRF mode (*Tafteh et al., 2016*). The results are displayed in *Figure 6A*.

Images of tetramer clusters near the cell surface were placed into an array with columns designating the treatment (buffer [Control]; 10 μmol/L FKBP12; 10 μmol/L FKBP12.6), and rows indicating whether the cells were (bottom) or were not (top) exposed to the phosphorylation cocktail. The images demonstrate dramatic, widespread, changes in the appearance and shape of the clusters. The immunophilins appeared to have fractured the clusters into smaller pieces, while all of the phosphorylated cells appeared to have larger clusters than their non-phosphorylated counterparts.

As demonstrated in *Figures 2H*, *3F* and *4*, the centre-to-centre NND between tetramers rarely exceeded 50 nm. We therefore defined a cluster in the dSTORM images as a group of lit pixels that had no pixel separated from its nearest neighbor by more than 50 nm. Each cluster had an alpha shape drawn around it enabling calculation of the enclosed area. The clusters were irregularly shaped and sometimes included subgroups (arrows, *Figure 6B*). We then pooled all of the areas from a given experimental group and plotted them as a CDF (*Figure 6C*). Regardless of the experimental treatment the areas varied in size by two orders of magnitude. The median cluster area and the number of clusters analyzed in each experimental group are listed in *Table 2*. The k-sample

**Table 1.** Tetramer Arrangements.

|  | Checkerboard | Side-by-Side | Isolated | Both | Tomograms | Tetramers | Rats |
|---|---|---|---|---|---|---|---|
| FKBP12 | 8.4% | 57.1% | 18.5% | 16.0% | 5 | 119 | 3 |
| FKBP12+Phos | 60.5% | 13.2% | 19.4% | 7.0% | 7 | 129 | 3 |
| FKBP12.6 | 8.2% | 82.8% | 7.4% | 1.6% | 7 | 122 | 3 |
| FKBP12.6+Phos | 71.3% | 5.2% | 22.6% | 0.9% | 8 | 115 | 3 |
| Control | 50.0% | 27.6% | 14.3% | 8.2% | 5 | 98 | 3 |
| Phosphorylation | 75.0% | 6.3% | 14.1% | 4.7% | 4 | 64 | 3 |

**Figure 3.** Localization, Distribution and NND Measurements of RyR2 on the Surface of the jSR in a Permeabilized Rat Ventricular Myocyte Exposed to the Phosphorylation Cocktail. Orthogonal planes within the tomogram are outlined in different colors; XY in red, YZ in green and XZ in blue. (A) A single XY plane extracted from the dual-tilt tomogram. Single arrow points to the jSR, and the double arrow points to the t-tubule. (B) YZ view. (C) (i) An enlarged area of the junction in A. (C) (ii) Multiple XZ planes, at 0.5 nm intervals, were positioned within the cleft to bisect the ryanodine receptors. The planes paralleled, as nearly as possible, the jSR membrane; two of which (planes 2 and 8) are highlighted in blue. (C) (iii) Demonstration of how the tilt in X changed the appearance of the tetramers in Cii8. (D) Identification of RyR2 in Cii8. Each required a different rotation to identify the tetramers' corners (red arrows). (E) Final distribution and orientation of all of the tetramers. The tetramers identified in Di and Dii are highlighted. (F) Histogram of the NND of 61 tetramers in four junctions.

Anderson Darling test indicated that the six distributions were not all drawn from the same population (p<0.01). The clusters produced by saturating concentrations of FKBP12 and 12.6 were not significantly different from each other, but were significantly smaller compared to control (p<0.01). Cells exposed to the phosphorylation cocktail, both with and without immunophilins, had significantly larger clusters than the control cells (p<0.01). Importantly, these results show that hyperphosphorylation takes precedence over excess FKBP12 or 12.6.

To quantify the dSTORM images, we used the higher resolution tomographic images to guide our analysis. We first drew alpha shapes around the tetramer arrays in the tomograms (*Figure 6D*) to calculate the median tetramer coverage/cluster for each of the experimental groups giving the percentage of the area covered by the tetramers; the values were different for the different experimental groups (*Table 3*). This is a measure of the group behavior of the

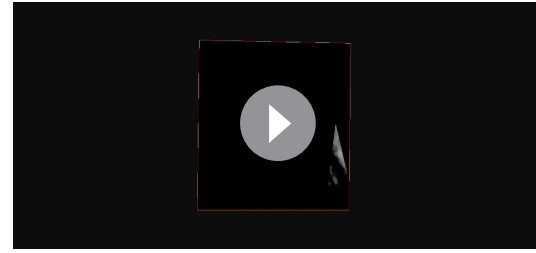

**Video 2.** Dyad from a Myocyte Treated with the Phosphorylation Cocktail: Tomogram and Model.
https://elifesciences.org/articles/51602#video2

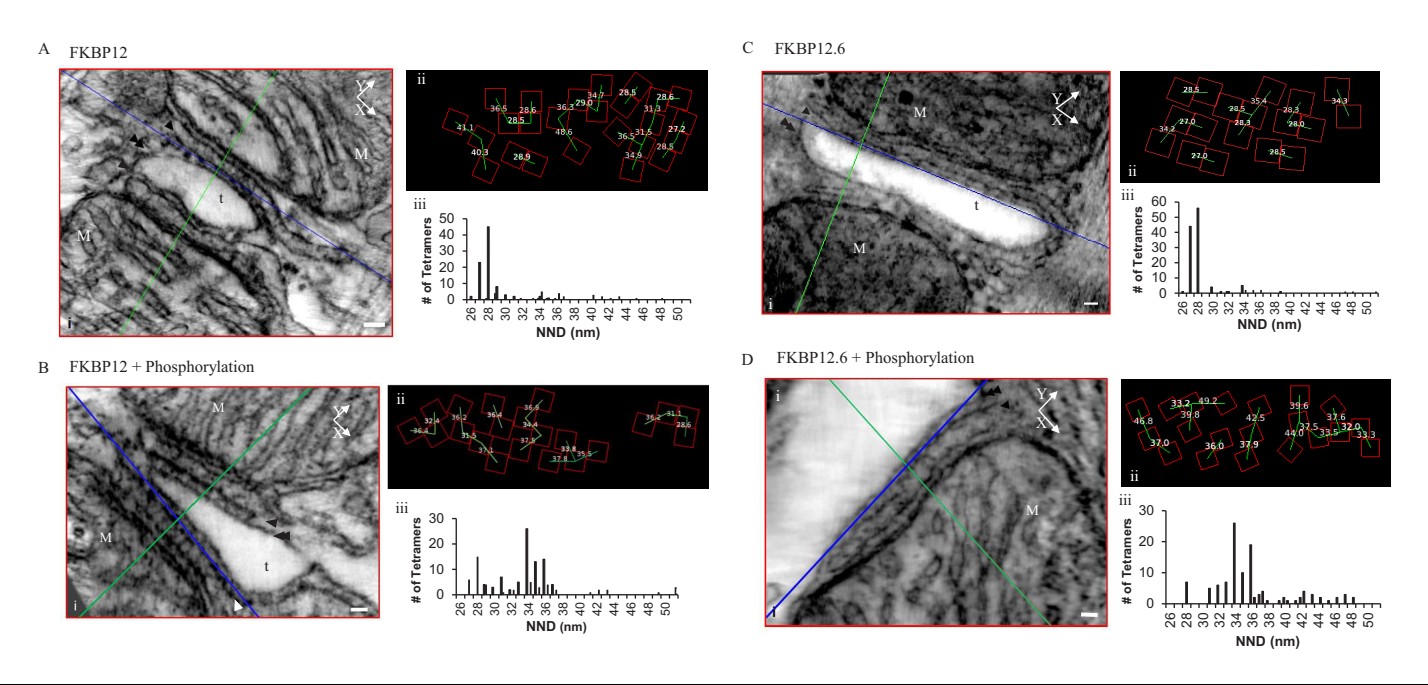

**Figure 4.** Permeabilized Rat Cardiomyocytes Treated with 10 µM Exogenous, Purified, FKBP12 or FKBP12.6 Either Without (**A and C**) or With (**B and D**) the Phosphorylation Cocktail. (i) A single XY plane (red) through the dyad with the YZ (green) and XZ (blue) planes intersecting a single tetramer. M – mitochondria, t- t tubule, scale bars 30 nm. (ii) Position, orientation and centre-to-centre NND of the tetramers. (iii) Histograms of the centre-to-centre NND of all of the tetramers in the given experimental group (*Table 1*).

array and is not comparable to the NND. The phosphorylated value was slightly larger than the control because ordering the array reduced the area of the alpha shape so that the tetramers occupied slightly more of the area defined. In the immunophilin treated cells the receptors clustered together reducing the alpha shape area and increasing the proportion covered by the tetramers. Phosphorylation of the immunophilin treated cells had the opposite effect – separating the tetramers, increasing the alpha shape area and decreasing the proportion covered. For each of the dSTORM images we multiplied the cluster area by coverage/cluster giving the area covered by the tetramers. When this was divided by the area occupied by a single tetramer, 729 nm (*Van Petegem, 2015*), the result was the number of tetramers per cluster in the dSTORM images (*Table 3*). The pooled values for each experimental group were plotted as a CDF, *Figure 6E*, where the steps in the curves reflected rounding to the nearest integer. The values shown, from two to ~100, are broadly consistent with published estimates (*Baddeley et al., 2009*; *Hayashi et al., 2009*). The k-sample Anderson Darling test showed that each curve was sampled from a different population (p<0.01). These quantitative results support the visual impression; the immunophilins fractured the clusters into smaller units

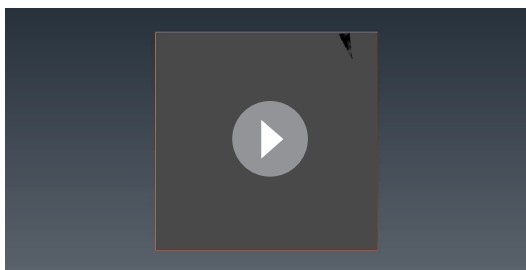

**Video 3.** Dyad of a Myocyte Treated with FKBP12: Tomogram and Model.
https://elifesciences.org/articles/51602#video3



**Video 4.** Dyad of a Myocyte treeated with FKBP12.6: Tomogram and Model.
https://elifesciences.org/articles/51602#video4

with fewer members. While phosphorylation, either with or without the immunophilins present, produced larger clusters with more members.

Median mean cluster areas and number of clusters examined. The alpha shapes and areas were generated using CGAL.

Median tetramer coverage per cluster recorded from the indicated number of tomograms using the RyR_fit program (*Source code 2*). Mean and median number of tetramers per cluster.

## Discussion

We have provided compelling evidence that inhibitory factors, such as the immunophilins, are associated with the RyR2 tetramers moving into a largely side-by-side configuration and decreasing the cluster size. In direct contrast, excitation due to phosphorylation, even in the presence of the inhibitory immunophilins, is associated with the tetramers moving apart into a checkerboard-like configuration and increasing the cluster size. These findings show that our understanding of RyR2 function has been incomplete and a full understanding should incorporate these dynamic effects. The results confirm and extend our earlier findings, and demonstrate that normal operation of the mammalian ventricular dyad involves movement and rearrangement of the tetramers and changes in the cluster size.

### Tetramer arrangement and cluster size

The distribution of nearest-neighbor centre-to-centre distances (NND) identified in control myocytes was bimodal, comparable to our previous results, with peaks at 28 and 34 nm. The median of the two modes varied little (*Table 4*) with a range of 1.4 nm for the checkerboard arrangement and 1.1 nm for the side-by-side arrangement. The modes indicate the preferred arrangements; the tetramers fully abut each other in the side-by-side arrangement while in the checkerboard configuration the adjacent turrets would overlap by 6–7 nm. However, we have observed a large range of NND values corresponding to tetramers that are in a wide variety of configurations; from fully side-by-side to completely isolated from each other. In a study of RyR2 deposited on a grid, Cabra et al reported unexpected tetramer interactions, suggesting that the tetramers can contact each other at multiple sites, not solely at the clamp domains as previously thought (*Cabra et al., 2016*).

Recently published superresolution data appeared to show mostly single and isolated tetramers in a unimodal distribution with a mean NND of 40 nm (*Jayasinghe et al., 2018*). In contrast, our data shows that almost all the tetramers appear in groups of two or more and are rarely separated by more than 39 nanometers, a distance corresponding to a corner to corner configuration. It is clear when examining our results that the distributions are not random and that the tetramers are not positioned independently of each other. This conclusion is also supported by the data in *Table 4* demonstrating the consistently preferred centre-to-centre NND of 34 nm in the checkerboard-like configuration and the large proportion of those that are side-by-side; results that are inconsistent with random or independent positioning. The reasons for these differences are unclear, but the resolution of electron tomography is an order of magnitude higher than any current optical approach and the tetramers are directly visualized, as opposed to indirectly via an antibody-dependent technique.

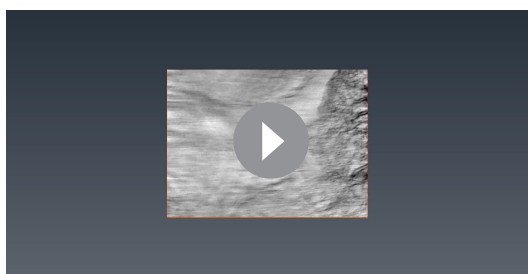

**Video 5.** Dyad of a Myocyte treated with FKBP12 and the Phosphorylation Cocktail: Tomogram and Model.
https://elifesciences.org/articles/51602#video5

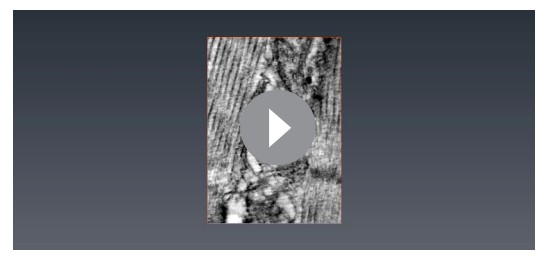

**Video 6.** Dyad of a Myocyte treated with FKBP12.6 and the Phosphorylation Cocktail: Tomogram and Model.
https://elifesciences.org/articles/51602#video6

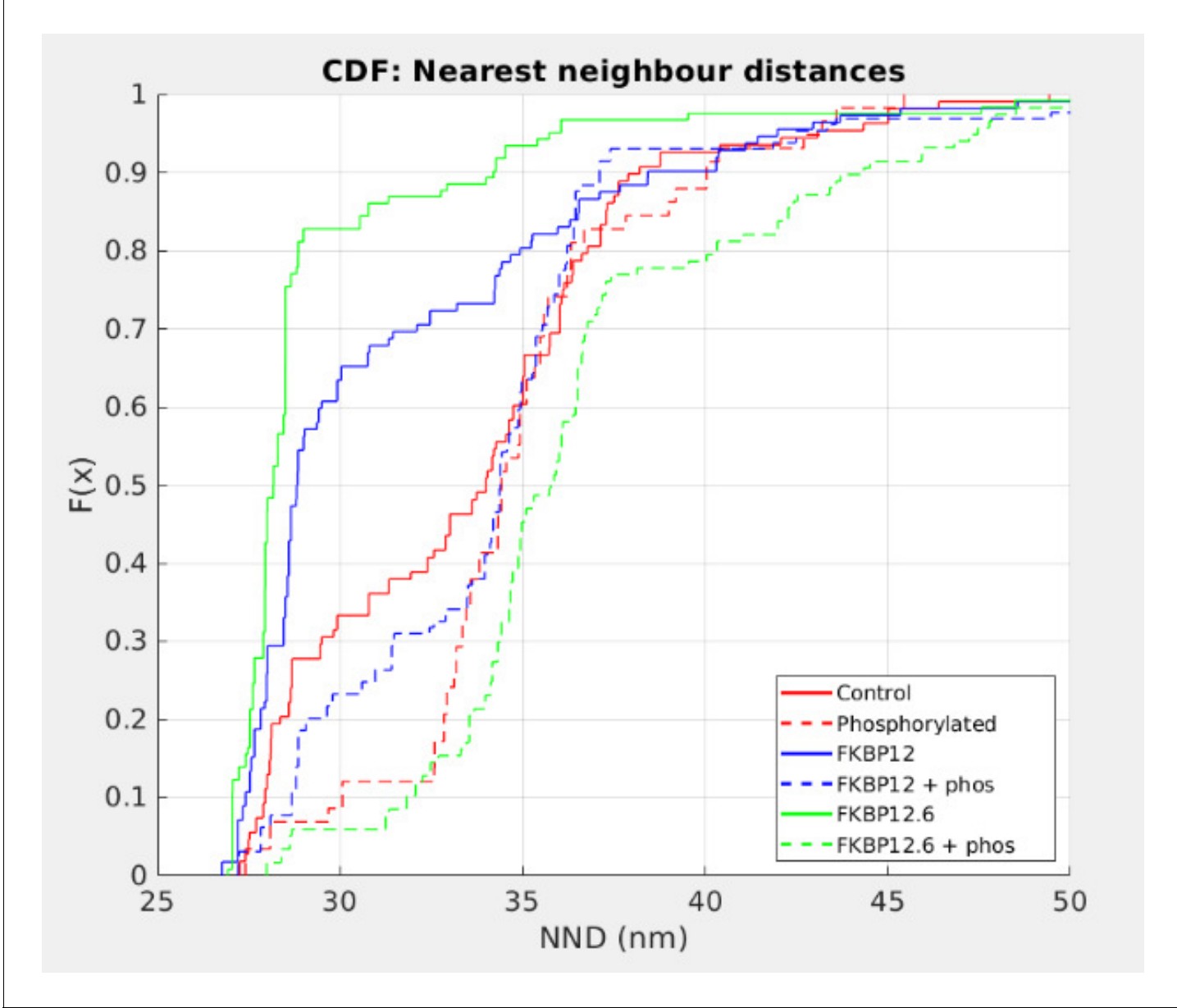

**Figure 5.** Cumulative Distribution Functions of the C-to-C NND. Each treatment group is associated with a different color and all are significantly different from each other (p<0.01). Left shifted lines have significantly smaller NND than control while those that are right shifted have NND that are significantly larger. Full statistics and the Matlab code are available in *Figure 5—source data 1* and *Source code 2* respectively.

The online version of this article includes the following source data for figure 5:

**Source data 1.** Statistical Analysis of the Cumulative Distribution Functions.

Both FKBP12 and FKBP12.6 produced dramatic shifts in the tetramers' relative positions, packing them into a predominantly side-by-side configuration, with FKBP12.6 being more effective than FKBP12 (*Figure 4* and *Table 1*). The results were comparable to those we reported for high $Mg^{2+}$, which also inhibits RyR2 (*Asghari et al., 2014*). dSTORM images of RyR2 clusters on cell surfaces demonstrated that the effects were widespread, as expected, but they also revealed other details.

*Figure 6*, and the data in *Tables 2* and *3*, show that saturating the tetramers with immunophilins resulted in clusters that occupied a smaller area and had fewer members, while phosphorylation, even in the continued presence of the immunophilins, did the opposite.The data supports RyR2 trafficking over significantly larger distances than we would have predicted from the tomographic

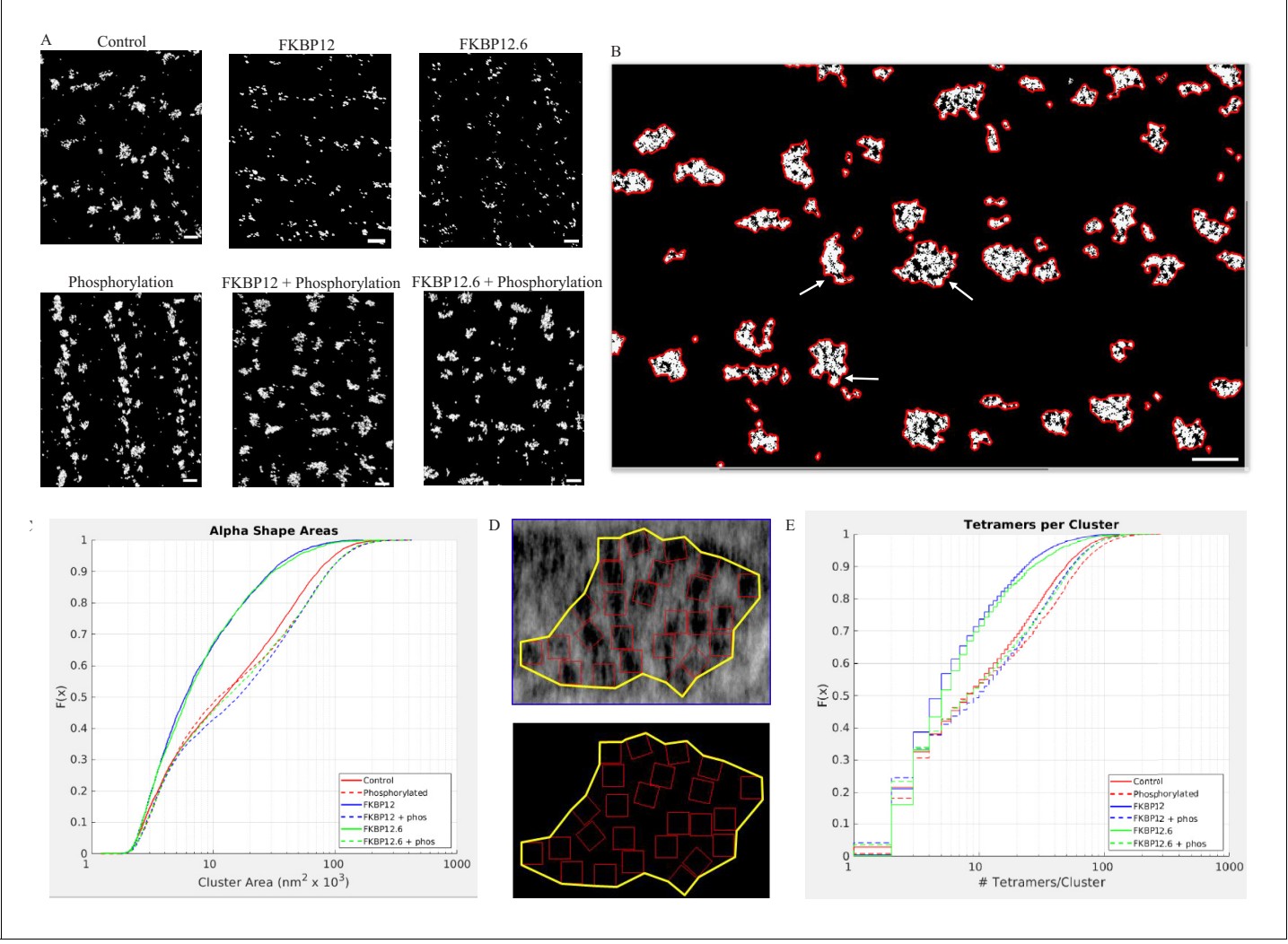

**Figure 6.** Superresolution Images of the Distribnution of RyR2 on the Myocyte Surface. (**A**) Superresolution images of the surface of isolated cardiomyocytes labeled for the ryanodine receptor under the indicated conditions. Scale bars are 500 nm. (**B**) Alpha shapes identify individual clusters. Arrows point to examples of subgroups within a cluster. Scalebar is 500 nm. (**C**) Cumulative distribution functions of the RyR2 cluster sizes. Each treatment group is associated with a different color. (**D**) Alpha shape drawn around a tetramer array, the tomographic images are from a control myocyte. (**E**) Cumulative distribution functions of the number of RyR2 within a cluster under the different experimental conditions. Full statistics and the Matlab code are available in *Figure 6—source datas 1* and *2* and *Source code 2*.

The online version of this article includes the following source data and figure supplement(s) for figure 6:

**Source data 1.** Statistical Analysis of the Cumulative Distribution Functions for the Alpha Shape Areas.
**Source data 2.** Statistical Analysis of the Cumulative Distribution Functions for the Tetramers per Cluster.
**Figure supplement 1.** Screen Shot of the Output from the Fourier Ring Correlation Program: Resolution = 18.6 nm.

images alone. This data is consistent with other results that have recently reported relatively rapid and unexpectedly long range trafficking of RyR2 clusters (*Hiess et al., 2018*) and individual tetramers (*Fu et al., 2016*).

## Ca²⁺sparks and FKBP12/12.6

The interpretation of our $Ca^{2+}$ spark data is dependent on an intact myocyte architecture, so we have compared our results to others who have used comparable experimental systems.

We demonstrated that FKBP12.6 reduced spontaneous SR $Ca^{2+}$ release and increased SR $Ca^{2+}$ content, which is broadly consistent with the published literature. Using permeabilized rat myocytes, *Guo et al. (2010)*. demonstrated that the application of 100 nmol/L fluorescent FKBP12.6 reduced

**Table 2.** dSTORM Images: Cluster Areas.

| | Median Cluster Area (nm² x 10³) | Mean Cluster Area (nm² x 10³) | Number of Clusters Examined |
|---|---|---|---|
| Control (10) | 12.2 | 27.3 | 5582 |
| Phosphorylated (12) | 11.1 | 32.2 | 4997 |
| FKBP12 (13) | 5.9 | 12.4 | 3615 |
| FKBP12.6 (7) | 6.1 | 13.2 | 2444 |
| FKBP12 + Phos (10) | 16.0 | 33.3 | 7761 |
| FKBP12.6 + Phos (9) | 13.1 | 32.3 | 4248 |

$Ca^{2+}$ release from the SR by reducing $Ca^{2+}$ spark frequency, amplitude and duration. Viral mediated overexpression of FKBP12.6 in both rat (*Gómez et al., 2004*) and rabbit myocytes (*Loughrey et al., 2004*; *Prestle et al., 2001*) reduced $Ca^{2+}$ spark frequency, amplitude and duration resulting in an increase in SR $Ca^{2+}$ content and larger $Ca^{2+}$ transients. These results are consistent with each other indicating an inhibition of RyR2 $Ca^{2+}$ release, and with those obtained from FKBP12.6 null mice, showing an elevated $Ca^{2+}$ spark width and duration (*Xin et al., 2002*) and an increase in the open probability of channels isolated from these animals (*Wehrens et al., 2003*).

In our experiments, FKBP12's effects on $Ca^{2+}$ sparks were comparable to those of FKBP12.6, but the published results are in conflict. *Guo et al. (2010)* found that fluorescent FKBP12 reduced the $Ca^{2+}$ spark amplitude and spread, but caused only a small, non-significant drop in spark frequency. The difference between their results and ours might be due to the higher RyR2 saturation we obtained using 10 µmol/L FKBP12 in comparison to their 1 µmol/L. Transgenic rabbit models acutely overexpressing FKBP12 showed a reduced $Ca^{2+}$ spark frequency and gain of ECC, but they displayed an increased $Ca^{2+}$ spark amplitude and duration. FKBP12 null mice largely die in utero and have severe dilated cardiomyopathy. In general, the data indicates that FKBP12 regulates RyR2, but the reasons for the conflicting results are unknown.

## Phosphorylation

Phosphorylation of RyR2 has been reported to reduce the affinity for FKBPs resulting in their dissociation and an increase in the tetramers' open probability and spark frequency; a phenomenon allegedly responsible for the increased diastolic $Ca^{2+}$ leak in heart failure (*Marx et al., 2000*). The finding is controversial as many laboratories have been unable to reproduce the results, and predictions from the hypothesis have failed to materialize (*Xiao et al., 2004*; *Guo et al., 2010*; *Alvarado et al., 2017*; *Xiao et al., 2007*).

Multiple lines of evidence support the conclusion that the immunophilins did not dissociate from RyR2. If they had, then results obtained from cells treated with the phosphorylation cocktail and cells treated with either FKBP12 or FKBP12.6 plus the phosphorylation cocktail would be equivalent. This was not the case in any of our experiments. The $Ca^{2+}$ sparks had significant differences in their frequency, FDHM, FWHM and mass (*Figure 1* and *Figure 1—source data 1*). The nearest neighbour

**Table 3.** Tetramer Density per Cluster.

| | Tomograms | | dSTORM Images | |
|---|---|---|---|---|
| | Median Tetramer Coverage/ Cluster | No. of Tomograms Examined | Median No. of Tetramers/ Cluster | Mean No. of Tetramers/ Cluster |
| Control | 49.5% | 13 | 8 | 18.5 |
| Phosphorylated | 52.8% | 9 | 8 | 23.3 |
| FKBP12 | 57.1% | 12 | 5 | 9.7 |
| FKBP12.6 | 62.5% | 9 | 5 | 11.3 |
| FKBP12 + Phos | 44.9% | 7 | 10 | 20.5 |
| FKBP12.6 + Phos | 46.2% | 8 | 8 | 20.5 |

**Table 4.** Nearest Neighbour Centre-to-Centre Distances.

| | Checkerboard | | Side-by-Side | |
|---|---|---|---|---|
| | Mean ± SD | Median | Mean ± SD | Median |
| FKBP12 | 34.8 ± 2.0 | 35.0 | 28.4 ± 0.9 | 28.5 |
| FKBP12 + Phos | 34.3 ± 1.9 | 34.6 | 28.6 ± 0.9 | 28.8 |
| FKBP12.6 | 33.3 ± 2.8 | 33.6 | 28.0 ± 0.8 | 28.0 |
| FKBP12.6 + Phos | 35.1 ± 2.1 | 34.9 | 28.4 ± 0.2 | 28.4 |
| Control | 34.2 ± 1.6 | 34.6 | 28.3 ± 0.8 | 28.1 |
| Phosphorylation | 34.2 ± 1.9 | 34.3 | 27.7 ± 0.3 | 27.7 |

distances between the tetramers were not equivalent (*Figure 5* and *Figure 5—source data 1*). In our dSTORM images there were significant differences in the areas of the clusters (*Figure 6C*, *Figure 6—source data 1* and *Table 2*) as well as in the number of tetramers per cluster (*Figure 6E*, *Figure 6—source data 2* and *Table 3*). Western blots showed an insignificant change in the intensity of the immunophilin bands following phosphorylation relative to their non-phosphorylated counterparts (*Figure 7* and *Figure 7—source data 1*).

These results are consistent with many others who have concluded that phosphorylation does not dissociate the immunophilins from RyR2. Co-immunoprecipitation experiments have demonstrated the continued binding of immunophilins to RyR2 despite phosphorylation by either CaMKII (*Kohlhaas et al., 2006*) or PKA (*Xiao et al., 2004*). Experiments with fluorescent FKBP12 and FKBP12.6 found no change in the $K_d$, $B_{max}$ or binding kinetics of the immunophilins in response to PKA activation (*Guo et al., 2010*). Finally, Western blots found no significant decrease in the intensity of the immunophilin bands following phosphorylation (*Jiang et al., 2002*; *Stange et al., 2003*; *Xiao et al., 2005*; *Xiao et al., 2006*).

Exposing immunophilin-treated tetramers to cell-wide phosphorylation largely reversed the immunophilins' effects, despite their remaining bound. The effects of phosphorylation plus FKBPs were much closer to phosphorylation alone than to FKBPs alone, making it likely that phosphorylation predominates over the FKBPs in determining the tetramers' function and that the FKBPs, phosphorylation and the combination of the two result in different conformations of RyR2. It is notable that phosphorylation was significantly less effective in reversing FKBP12's effects on spark frequency, FDHM and spark mass compared to FKBP12.6, despite its significantly lower affinity for RyR2 (*Figure 1B*). $Ca^{2+}$ sparks in cells treated with FKBP12.6 plus the phosphorylation cocktail were significantly more frequent than in any other treatment group (*Figure 1A and B*) and when combined with the increase in spark mass, these cells would have the largest diastolic $Ca^{2+}$ release.

## Differential Effects of FKBP12 and 12.6

The different effects of the FKBPs, with or without phosphorylation, occur despite their binding to the same site on RyR2 (*Zissimopoulos et al., 2012*). A possible explanation is that each immunophilin produces direct, unique, effects on RyR2 gating. Unfortunately, the data published from lipid bilayer studies is perplexing and neither proves nor disproves this contention. For example, FKBP12.6 has been shown to inhibit RyR2 (*Marx et al., 2000*), to have no effect (*Xiao et al., 2007*; *Barg et al., 1997*) and to act as a partial agonist (*Galfré et al., 2012*). FKBP12 has been reported to have no effect (*Barg et al., 1997*) and to increase RyR2 open probability (*Galfré et al., 2012*). Although a substantial part of the FKBP12 and 12.6 surfaces are involved in binding the SPRY1 and solenoid domains of RyRs, their outer surfaces are accessible in all current cryo-EM studies, and these may be directly involved in interactions with a neighboring RyR2. Since these sequences are less conserved, the contact sites might differ, resulting in different tetramer distributions and spark characteristics for the FKBPs. Finally, proteomic data may offer a different explanation, FKBP12 can be phosphorylated (*Lundby et al., 2012*), while no such reports exist for FKBP12.6. The functional consequences of RyR2 saturated with FKBP12 versus phosphorylated FKBP12 are unknown.

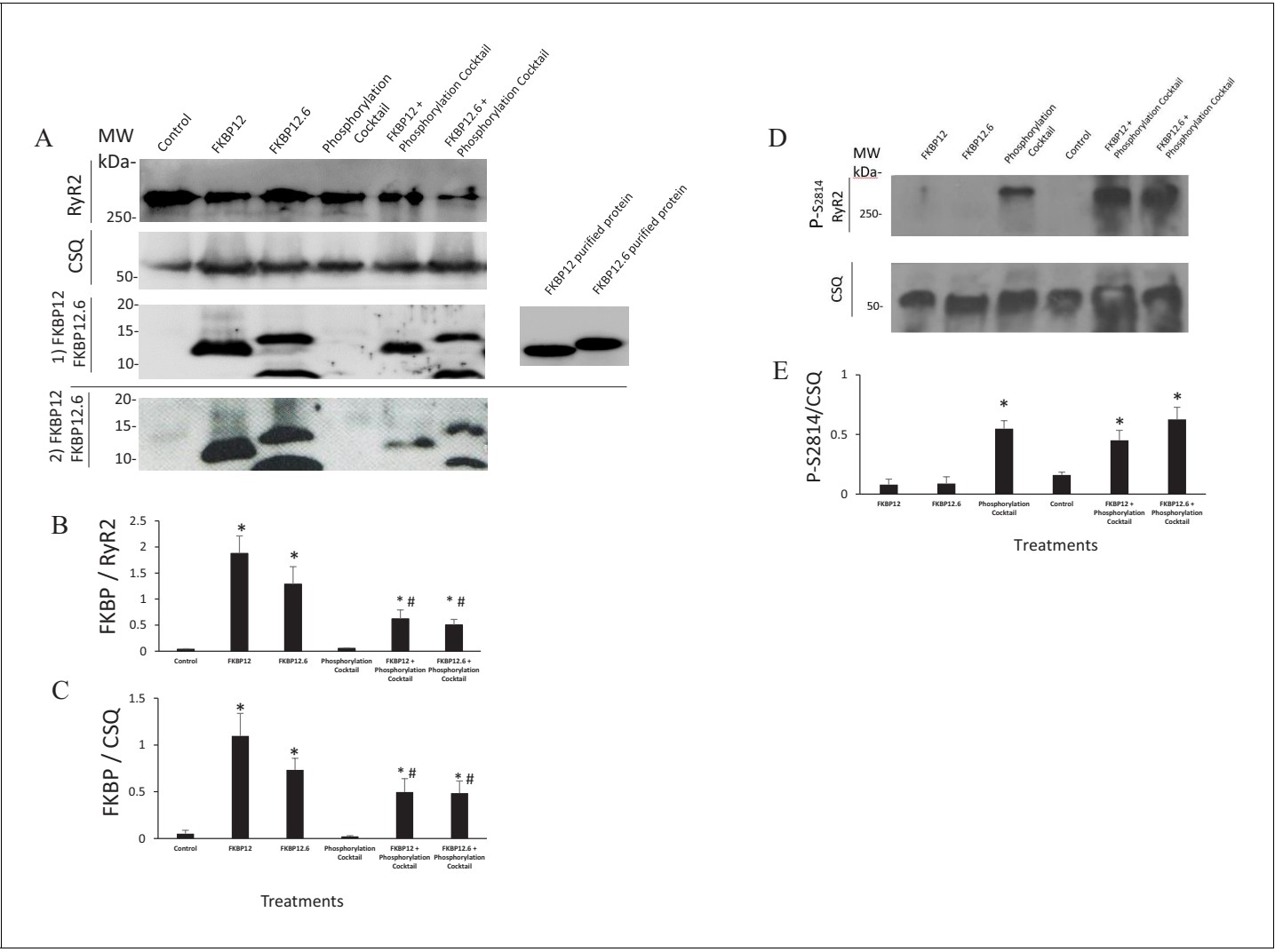

**Figure 7.** Western Blots. (**A**) Western blots probed with anti FKBP12 (two blots displayed, the bottom blot shows a visible band in control lane) and the proteins selected as loading controls, calsequestrin and RyR. (**B and C**) An average of the western blots demonstrated that FKBP12 and FKBP12.6 purified peptides produced a significant increase in FKBPs association with RyR2 in treated cells. (**D**) Western blots probed with anti p-S2814, a phosphorylation site on RyR2, and the loading control, calsequestrin (CSQ). (**E**) An average of the blots demonstrated that the phosphorylation cocktail produced a significant increase in phosphorylation of S2814. Band densities were measured using ImageJ and were expressed as mean ± SEM. n = 7 blots of 4 hearts in B and C. n = 4 blots of 4 hearts in E. Data were analyzed using student t-test. *p<0.05 vs control. #p<0.05 vs Phosphorylation cocktail.

The online version of this article includes the following source data for figure 7:

**Source data 1.** Statistical Analysis of the Western Blots.

## Structure and function

Sparks are formed by the spontaneous opening of one or more RyR2 channels, which open their nearest neighbors via $Ca^{2+}$ diffusion and inter-RyR2 CICR within a single dyadic cleft. Intuitively, inter-RyR2 should be maximally effective and spark frequency the highest when the channels are closest together. This is reflected in some models of spark formation that predict the closer the tetramers are to each other the higher the probability of spark formation, while the farther apart, the less likely its occurrence (*Walker et al., 2015*). Our results demonstrate a contrary relationship: all of the factors that redistribute RyR2 into a more compact distribution, high $Mg^{2+}$ (*Asghari et al., 2014*) as well as FKBP12 and FKBP12.6, result in a significantly lower spark frequency than control, while phosphorylation, whether or not an immunophilin is present, separates the tetramers and

increases the spark frequency above its control values. dSTORM imaging has reinforced and extended these observations. Visible changes in the organization of the RyR2 clusters in each of the images (*Figure 6A*) demonstrated that the effects of our experimental manipulations were wide-spread. The quantitative analysis reinforced those impressions showing highly significant changes in the sizes of the clusters in response to our experimental treatments (*Figure 6C*). The data implies that tetramers were being translated over larger distances than anticipated from the tomographic images. Notably, the changes in cluster size were correlated with significant changes in the $Ca^{2+}$ spark parameters. Experimental manipulations that reduced $Ca^{2+}$ spark frequency and related parameters also reduced the number of tetramers per cluster, while those that increased the cluster sizes had the opposite effect (*Figures 1* and *6*, and *Table 3*).

We have speculated that the changes necessary for channel opening are inhibited when the tetramers' sides are overlapped and in contact (*Asghari et al., 2014*). Subsequently, high resolution structural analyses have demonstrated that the cytoplasmic shell of the RyR2 molecule undergoes significant reorientation and movement on channel opening which could support this hypothesis (*des Georges et al., 2016*).

Recent data demonstrates that the probability of $Ca^{2+}$ spark formation is related to the number of tetramers in the cluster, with larger clusters being more likely to produce a $Ca^{2+}$ spark (*Galice et al., 2018*; *Xie et al., 2019*). Our data indicates a comparable relationship, but is the first demonstration that cluster size is a variable and might be regulated by factors that affect tetramer gating. This suggests that the movement and translation of the tetramers we have observed is a mechanism to alter the probability of $Ca^{2+}$ spark formation.

The changes in the tetramer arrangement and the cluster size both occur relatively rapidly, within ten minutes. It has been shown that BIN1 is involved both in the formation of $Ca^{2+}$ release units and in anchoring the jSR (*De La Mata et al., 2019*) as well as in the rapid translocation of RyR2 onto the jSR membrane in response to β-adrenergic stimulation (*Fu et al., 2016*). It could therefore contribute to the tetramer movements and dyad reorganization we have observed. Whether both the short range tetramer reorientation occurring within a dyad, and the longer range translation to and from a dyad, occur sequentially, in synchrony or independently is unknown, and they may involve different mechanisms. The clear implication is that the mammalian dyad is a dynamic environment that can rapidly change, resulting in effects on excitability, diastolic $Ca^{2+}$ release and contractility.

## Materials and methods

**Key resources table**

| Reagent type (species) or resource | Designation | Source or reference | Identifiers | Additional information |
|---|---|---|---|---|
| Biological sample (Rat) | Primary left ventricular myocytes | Charles River Canada | RRID: RGD_11040548 | Wistar Male rat (150–200 gr) |
| Antibody | Anti FKBP12 (Rabbit polyclonal) | Abcam | RRID: AB_303413 | WB (1:2500) |
| Antibody | Anti Calsequestrin (Rabbit polyclonal) | Abcam | RRID: AB_303865 | WB (1:75000) |
| Antibody | RYR2 Phospho Serine 2814 Anti-Serum antibody (Rabbit polyclonal) | Badrilla | RRID: AB_2617055 | WB (1:2000) |
| Antibody | Ryanodine Receptor Monoclonal Antibody (C3-33) | Thermo Fisher Scientific | RRID: AB_2183054 | Super-res IF(1:100), WB (1:1000) |
| Antibody | Goat anti-Mouse IgG (H+L) Highly Cross-Adsorbed Secondary Antibody, Alexa Fluor 647 | Thermo Fisher Scientific | RRID: AB_2535805 | Super-res IF (1:50) |
| Antibody | Anti-Rabbit IgG (H+L) Polyclonal Antibody, Horseradish Peroxidase Conjugated | Innovative Research (Invitrogen/Zymed) | RRID:AB_88349 | WB (1:10000) |

*Continued on next page*

*Continued*

| Reagent type (species) or resource | Designation | Source or reference | Identifiers | Additional information |
|---|---|---|---|---|
| Antibody | Goat anti-Mouse IgG (H+L) Secondary Antibody, HRP | Thermo Fisher Scientific | RRID:AB_2533947 | WB (1:10000) |
| Other | Immobilon-P PVDF membrane | Sigma Millipore | Cat# IPVH00010 | WB (0.45 µm pore size) |
| Other | FluoSpheres Carboxylate-Modified Microspheres, 0.1 µm, infrared fluorescent | Thermo Fisher Scientific | Cat# F8799 | Microspheres used in Super-res IF (Image stabilization) |
| Other | SuperSignal West Femto Chemiluminescent Substrate kit | Thermo Fisher Scientific | Cat# 34095 | Horseradish peroxidase (HRP) substrates- WB |
| Other (Calcium Indicator) | Fluo-4, Pentapotassium Salt | Thermo Fisher Scientific | Cat# F14200 | Concentration: 30 µmol/L |
| Chemical compound, drug | Protease Inhibitor Cocktail | Sigma | Cat#P-8340 | WB- lysis buffer |
| Chemical compound, drug | IBMX (3-isobutyl-1-methylxanthine/) | Sigma | Cat# 15879 | In phosphorylation cocktail as a phosphodiesterase (PDE) inhibitor (10 µmol/L) |
| Chemical compound, drug | Calyculin A from Discodermia calyx | Sigma | Cat# C5552 | In phosphorylation cocktail as an inhibitor of serine-threonine protein phosphatase 2A |
| Chemical compound, drug | Okadaic acid ammonium salt from Prorocentrum concavum | Sigma | Cat#O8010 | In phosphorylation cocktail- Inhibitor of type 1 and type 2A protein phosphatases |
| Chemical compound, drug | 2-mercaptoethanol | Sigma | Cat#M3148 | Super-res IF (140 mmol/L) |
| Chemical compound, drug | Glucose Oxidase from Aspergillus niger | Sigma | Cat#G0543 | Super-res IF (0.5 mg/ml) |
| Chemical compound, drug | Catalase from bovine liver | Sigma | Cat#C3155 | Super-res IF (40 µg/ml) |
| Software, algorithm | Hierarchical statistical data analysis | *Sikkel et al., 2017* | PMID: 29016722 | Used for statistical analysis of calcium spark parameters |
| Software, algorithm | Amira for Life Sciences | Thermo Fisher Scientific (FEI) | RRID: SCR_007353 | Tomography 3D visualization and analysis software |
| Software, algorithm | ImageJ | NIH Image | RRID: SCR_003070 | Used in WB analysis |
| Software, algorithm | Fiji | NIH Image | RRID: SCR_002285 | Spark Master plugin, used in calcium spark analysis |
| Transfected construct (include species here) | *E. coli* Rosetta TM (DE3) cells | Novagen | Cat#70954 | Expression and purification of human FKBP12 and FKBP12.6 |
| Other | Superdex 200 16/60 column | GE Healthcare | Cat#GE28-9893-35 | Expression and purification of human FKBP12 and FKBP12 |
| Other | PorosMC column | Thermofisher Scientific | Cat#1542226 | Expression and purification of human FKBP12 and FKBP12 |

*Continued*

| Reagent type (species) or resource | Designation | Source or reference | Identifiers | Additional information |
|---|---|---|---|---|
| Chemical compound, drug | lysozyme | Thermofisher Scientific | Cat#89833 | 25 µg/ml (Expression and purification of human FKBP12 and FKBP12) |
| Chemical compound, drug | TEV protease | NEB | Cat#P8112S | Expression and purification of human FKBP12 and FKBP12 |
| Chemical compound, drug | imidazole | Sigma | Cat#792527 | 300 mmol/L (Expression and purification of human FKBP12 and FKBP12) |
| Chemical compound, drug | isopropyl-β-D-thiogalactoside (IPTG) | Thermofisher Scientific | Cat#BP-1755 | 0.4 mmol/L (Expression and purification of human FKBP12 and FKBP12) |
| Chemical compound, drug | PMSF | Sigma Millipore | Cat#7110 | 100 µM (Expression and purification of human FKBP12 and FKBP12) |
| Chemical compound, drug | DNase I | Sigma | Cat#11284932001 | 25 µg/ml (Expression and purification of human FKBP12 and FKBP12) |

The experiments used ventricular myocytes from adult rats. Animal handling was done in accordance with the guidelines of the Canadian Council on Animal Care and approved by the animal research committee of the University of British Columbia (UBC). All chemicals were purchased from Sigma-Aldrich (Oakville, ON) unless otherwise stated.

## Correlation microscopy

Isolation of rat ventricular myocytes was performed as previously described (*Rodrigues and Severson, 1997*). Isolated, permeabilized myocytes were exposed to one of (10 min exposure each): a) buffer (as control), b) phosphorylation cocktail, c) 10 µmol/L FKBP12.6, d) 10 µmol/L FKBP12.6, then the phosphorylation cocktail, e) 10 µmol/L FKBP12, f) 10 µmol/L FKBP12, then the phosphorylation cocktail. The phosphorylation cocktail was composed of (µmol/L): 10 c-AMP, 10 IBMX, 10 Okadaic acid and 0.5 Calycullin A (pH7.2; adjusted with KOH) (*Asghari et al., 2014*). $Ca^{2+}$ sparks were recorded on a Zeiss LSM 700 confocal microscope in line scan mode (63x objective) using 30 µmol/L Fluo-4 (Invitrogen). Sparks were analyzed using SparkMaster (*Picht et al., 2007*). $Ca^{2+}$ spark frequency was corrected for SR $Ca^{2+}$ content to isolate effects on the array from those due to changes in SR filling (*Li et al., 2002*). Spark duration and spread were recorded as full duration half maximum (FDHM; ms) and full width half maximum (FWHM; µm). Cells were then fixed, in place, and prepared for electron microscopy. A gridded coverslip enabled tomograms to be acquired from cells whose $Ca^{2+}$ sparks had been recorded. Groups of cells prepared and treated identically were exposed to 20 mmol/L caffeine to assess SR $Ca^{2+}$ content. The experimental pipeline for this novel correlative light and electron microscopy technique is detailed in *Figure 8*.

Measurements of FKBP12 and FKBP12.6 concentration in rat myocytes have produced wildly different results: (i) 1.8 FKBP12.6/RyR2 (45% occupancy) and no FKBP12 (*Zissimopoulos et al., 2012*); (ii) At most 20% of the subunits occupied by FKBP 12.6 and many of the rest by FKBP12 (*Guo et al., 2010*); (iii) FKBP12/RyR2 = 1.75 and FKBP12.6/RyR2 = 3.5 (*Jeyakumar et al., 2001*). Given these uncertainties we used high concentrations, 10 µmol/L of FKBP12 and FKBP12.6, to displace the endogenous proteins and to ensure that all four binding sites on each RyR2 were occupied with the desired isoform. We avoided rapamycin because recent experiments demonstrated that it may not displace FKBP12, only partially displace FKBP12.6, and may have direct effects on RyR2 by depressing both the $Ca^{2+}$ spark frequency and spread (*Gonano and Jones, 2017*; *Guo et al., 2010*; *Richardson et al., 2017*).

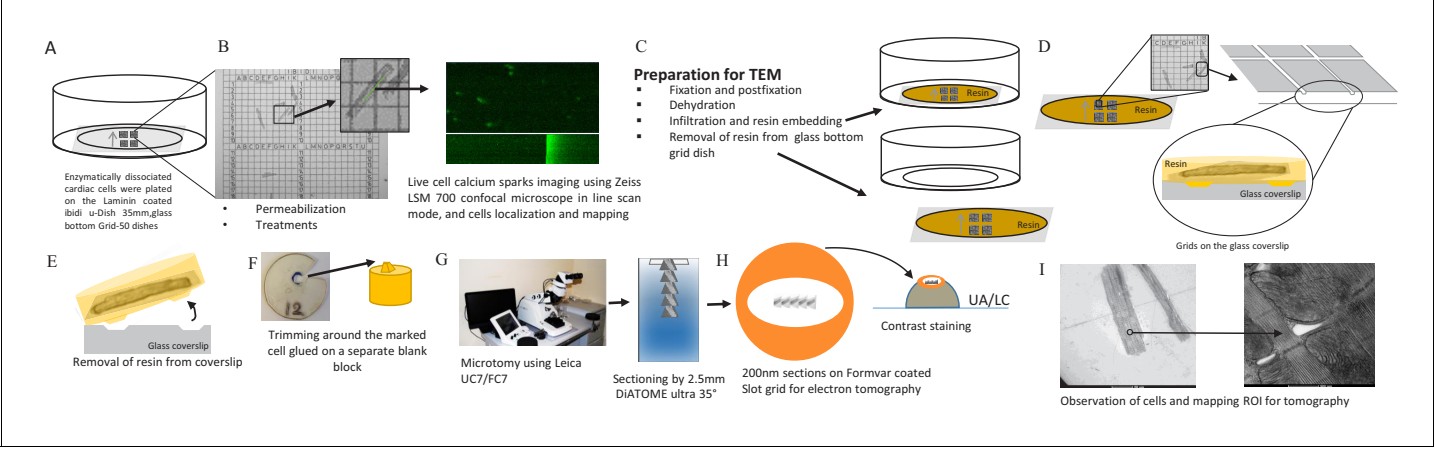

**Figure 8.** Calcium Spark and TEM Correlation Microscopy Pipeline. (**A**) Enzymatically dissociated cardiac cells were plated on laminin coated ibidi u-Dish 35 mm, glass bottom Grid-50 dishes (Laminin, Roche). (**B**) Permeabilized cardiac myocytes were exposed to fluorescent Ca2+ indicators (30 μmol/L Fluo-4) and to intracellular solution (IS) or to IS plus the phosphorylation cocktail for 10 min. Calcium sparks were recorded on a Zeiss LSM 700 confocal microscope in line scan mode (63/1.4 Plan Apochromat objective). Cells were chosen for calcium spark imaging and their location on the gridded dish was recorded. (**C**) Cells were prepared (fixed, dehydrated and resin embedded) for TEM in the grid dish. (**D**) The glass mesh coverslip was removed from the petri dish. (**E**) Resin was peeled off the glass coverslip using liquid nitrogen (details are in the Materials and methods). (**F**) The resin was trimmed and glued onto a separate blank resin block. (**G**) Using a Leica UC7/FC7 microtome, the blocks were sectioned on Fomvar coated Slot grids using a 2.5 mm DiATOME ultra 35° knife. (**H**) Contrast staining was done using the classic contrast method; slot grids were placed on a drop of uranyl acetate and then lead citrate. (**I**) The cells were observed by TEM and ROI were mapped for tomography.

The online version of this article includes the following figure supplement(s) for figure 8:

**Figure supplement 1.** Dual Tilt Electron Tomography; Imaging and Data Processing Pipeline.
**Figure supplement 2.** 2D Electron Micrographs of Treated Isolated Cells.

## Electron tomography

Cells were examined on a dual-tilt electron microscope, a 200 kV Tecnai G2 (FEI, Hillsboro, OR) using our established protocols. We used Inspect 3D to align and Amira 5.3 (VSG, Burlington, MA) to visualize the data sets, and employed no image processing steps other than a contrast stretch (*Figure 8—figure supplement 1*). 2D electron micrographs were taken of the cells after they had been permeabilized, $Ca^{2+}$ sparks had been recorded, and they were fixed, embedded, sectioned and ready for tomographic analysis The 2D EM images demonstrate that the ultrastructure was well preserved regardless of the experimental protocol (*Figure 8—figure supplement 2*). Additional details of cell preparation and acquisition are provided in *Asghari et al. (2014)*.

## Classifying RyR2 positions

An individual tetramer's position relative to its neighbors was classified using Amira and a custom-written C++ program, 'RyR_fit', as previously described (*Asghari et al., 2014*). Briefly, a tetramer was in a checkerboard arrangement relative to its neighbor(s) if the sides were parallel, separated by less than 3 nm and overlapped by less than 18 nm (2/3 of its length). If those criteria were fulfilled but the overlap exceeded 18 nm, the tetramers were classified as side-by-side. Some tetramers had neighbors in both configurations, while others had none and were considered isolated. The centre-to-centre nearest neighbor distances (NND) for the tetramers were recorded and plotted. The error in placing the tetramer is ~3 nm. The basis of this estimate, and other possible errors, is discussed in detail in *Asghari et al. (2014)*.

## Expression and purification of human FKBP12 and FKBP12.6

FKBP12 and FKBP12.6 were cloned to obtain a fusion protein with N terminal histidine and maltose binding protein (MBP) tag. FKBPs were induced in *E. coli* Rosetta (DE3) cells with 0.4 mmol/L isopro-pyl-β-D-thiogalactoside (IPTG) at 0.6 optical density at 600 nm. After growth for 4 hr at 37°C, the cells were collected, re-suspended in the buffer A (20 mmol/L HEPES pH 7.4, 250 mmol/L NaCl) with 100 μM PMSF, 25 μg/ml DNaseI and 25 μg/ml lysozyme and 10% glycerol, and disrupted by

sonication. Cell debris was removed by centrifugation at 40,000 g for 30 min. Cell lysate was applied to a PorosMC column (Thermo Fisher Scientific), were washed with five column volumes (CV) of buffer A and 5 CV of buffer A plus 10 mmol/L imidazole, and were eluted with buffer A plus 300 mmol/L imidazole [pH 7.4]). The protein was dialyzed overnight against buffer A and was cleaved simultaneously with recombinant TEV protease. The samples were then run on another PorosMC column in buffer A, and the flow-through was collected and dialyzed against buffer C (10 mmol/L NaCl, 5 mmol/L 2-mercaptoethanol, 20 mmol/L Tris-Cl [pH 8.8]), applied to a Q Sepharose column (GE Healthcare), and eluted with a gradient from 0% to 50% buffer D (1 M NaCl, 2-mercaptoethanol, 20 mmol/L Tris-Cl [pH 8.8]). Finally, the samples were run on a Superdex 200 16/60 (GE Healthcare) column in buffer A plus 5 mmol/L 2-mercaptoethanol, which separated dimeric and monomeric fractions. The monomeric fractions were pooled, aliquoted and flash-frozen for further use.

## Modeling

To visualize and better understand the 3D architecture and arrangement of the t-tubules, jSR and ryanodine receptors, we manually segmented these compartments on the dual tilt tomograms using Amira (FEI). Using the segmentation module, the contours were hand drawn, creating a 3D model, by tracing the individual membranes in every slice of the tomograms using a Wacom Cintiq 27QHD drawing tablet. In each dyad, a single RyR2 tetramer was traced, then copies of it were placed in the proper position and orientation; this simplified the drawing. As displayed in the supplementary videos, t-tubules were traced in blue, jSR in red and RyR2 tetramers in green. Automatic smoothing was applied to the surface models to reduce the number of polygons and improve the models' appearance. When drawing, the tomogram cannot be rotated, so to ensure that we identified the correct position and orientation of the tetramers we used the Slice option to superimpose the tomographic data onto the model. This simultaneous visualization ensured accurate positioning of the tetramers in the 3D data set. Movies of the complete models were created using Amira's MovieMaker module.

## Western blots

We used a standard Western Blot protocol to confirm that exogenous FKBP12 or FKBP12.6 raised the cellular levels beyond control values (*Figure 7*). The 6X sample buffer included: 350 mmol/L Tris-Cl (pH 6.8), 30% glycerol, 10% SDS, 600 mmol/L dithiothreitol, NaF (6 mmol/L) and a protease inhibitor (P-8340, Sigma) was directly added to the cells before (as a control) and after treatments. The samples were frozen in liquidN$_2$ for storage at −80°C until use. Protein (quantified with a Bradford Protein assay, Bio-Rad 500–0001, Hercules, CA) was separated on 12% SDS-polyacrylamide gels and transferred to Immobilon-P transfer membrane (PVDF – pore size 0.45 μm; Millipore Sigma) overnight at 4°C. Membranes were then blocked in skim milk in Tris-buffered saline with 0.1% Tween (TBS-T Sigma) at room temperature for one hour. Primary antibody incubation was performed at 4°C overnight and secondary antibody incubation at 37°C for one hour. Protein bands were visualized using the SuperSignal West Femto Chemiluminescent Substrate kit (Thermo Scientific, Rockford, IL).

We used the following antibodies: Anti-FKBP12 (ab2918; Abcam, Cambridge, UK) to detect both FKBP 12 and FKBP12.6, anti Calsequestrin (ab3516; Abcam) and anti-RyR2 (clone C3-33, MA3-916; Invitrogen, Carlsbad CA) were used as loading controls. Secondary antibodies: HRP-Goat anti-mouse IgG (H+L, 62–6520; Invitrogen) and HRP-Goat anti-rabbit IgG (H+L, 62–6120; Invitrogen).

## Statistical analyses

Since the spark data were not normally distributed, they are presented as a box and whisker plot. Parameters derived from the spark data were analyzed using *Sikkel et al. (2017)* method of hierarchical data analysis to remove the effects of variations in both the cells and the rats from which they were isolated. While spark frequency/Ca$^{2+}$ load was normally distributed, virtually none of the other parameters were, so, similar to Sikkel et al., we used the log transformation to analyze the data. We treated the p value generated by the R-program differently from Sikkel et al, using the Holm-Bonferroni correction (*Holm, 1979*) rather than the Bonferroni as the latter is statistically weaker and a higher family-wise error rate (*Olejnik et al., 1997*). The tables (*Figure 1B*, *Figure 1—source data 1*) describing the statistical results for each parameter have five columns – first the comparison, then the p values and the statistical significance generated by Sikkel et al's R program. The final two

columns show the results from the Holm-Bonferroni analysis, which involves ranking the probability from smallest to largest. A critical p value (column 4) is calculated according to the rank of the probability. If the p value (column 2) is less than this critical value, the null hypothesis is rejected (column 5).

The distribution of centre-to-centre NND of each of the groups was converted to a cumulative distribution function and then compared using the non-parametric k-sample Anderson-Darling test (*Scholz and Stephens, 1987*; *Scholz and Zhu, 2016*) to determine whether the distributions were significantly different from each other.

## Super-resolution immunofluorescence microscopy

The isolation of the cardiomyocytes, the labeling of the ryanodine receptor and the custom-built microscope have been described in detail elsewhere (*Tafteh et al., 2016*), but is notable for providing a highly stable imaging platform with drift in x and y of less than 0.7 nm RMS. Briefly, cells treated with the various cocktails as described in the experimental protocol were fixed, settled on coverslips and then labeled with anti-RyR2 and with anti-mouse Alexa 647 (Life Technologies, Burlington, ON). A low concentration of 100 nm tracking beads (F8799; Life Technologies) were added to the coverslips and allowed to settle overnight. Imaging was performed in a standard GLOX-thiol solution (a nitrogenated TN buffer [50 mmol/L Tris, 10 mmol/L NaCl, pH 8.0], 0.5 mg/ml glucose oxidase, 40 µg/ml catalase, 10% (w/v) glucose and 140 mmol/L 2-mercaptoethanol). The coverslip with the attached cardiomyocytes was mounted onto a chamber with a volume of 700 µl which was filled with the imaging solution and sealed. Blinks with an estimated error in their XY position >10 nm, Z position >40 nm, a goodness of fit <0.9, and nearest neighbors > 30 nm distant were excluded. An analysis of our images using Fourier ring correlation (*Banterle et al., 2013*) gave a resolution of between 16 and 25 nm. A sample result is shown in *Figure 6—figure supplement 1*. All superresolution images were displayed as binary; the pixel was on if blinks were present and off it was not.

To ensure that we could get an accurate measure of the area covered by the tetramer clusters, we imaged the surface of the cells where they contacted the coverslip and the tetramers would be near normal to the optical axis. A 100–150 nm thick slice that contained the surface and excluded structures beneath it was mapped onto a 2D grid of 10 nm x 10 nm pixels; those pixels that contained a single blink and had no neighbors were treated as empty.

To interpret the images, we defined a cluster as consisting of pixels where no pixel was more than 50 nm distant from its nearest neighbor. The area of each cluster was determined by defining the boundary with an alpha shape (*Bernardini and Bajaj, 1997*). Clusters with an area of less than 1600 nm (*Van Petegem, 2015*) were omitted from the analysis to remove noise that resulted from fluorophores which had attached to the coverslip. This filter also removed isolated single tetramers more than 50 nm distant from any cluster. We used data obtained from the tomograms to estimate the number of tetramers in a cluster. For each tomogram, we positioned the tetramers and then drew an alpha shape around the tetramers. Knowing the area occupied by a single tetramer (729 nm *Van Petegem, 2015*) we could calculate how much of the cluster was covered by the tetramers (median value, expressed as a percentage, *Table 3*). We then used these values to estimate the number of tetramers for each cluster. These values were plotted on as a cumulative data function (*Figure 6*) and were analysed using a k-sample Anderson-Darling test. Data was analyzed with programs using the Computational Geometry Algorithms Library (cgal.org).

## Acknowledgements

FVP acknowledges a grant from the CIHR (PJT-153305). This study was partially funded by the Natural Sciences and Engineering Research Council of Canada (KCC) and the Canada Foundation for Innovation (KCC). EDWM acknowledges a grant from CIHR (148527).

# Additional information

## Funding

| Funder | Grant reference number | Author |
| --- | --- | --- |
| Canadian Institutes of Health Research | 148527 | Edwin DW Moore |
| Canadian Institutes of Health Research | PJT-153305 | Filip van Petegem |
| Natural Sciences and Engineering Research Council of Canada | | Keng C Chou |
| Canada Foundation for Innovation | | Keng C Chou |

The funders had no role in study design, data collection and interpretation, or the decision to submit the work for publication.

## Author contributions

Parisa Asghari, Conceptualization, Investigation, Formal analysis, Supervision, Validation, Visualization, Methodology, Project administration; David RL Scriven, Data curation, Conceptualization, Software, Formal analysis, Visualization, Methodology, Investigation, Validation; Myles Ng, Formal analysis; Pankaj Panwar, Filip van Petegem, Resources; Keng C Chou, Resources, Software; Edwin DW Moore, Conceptualization, Resources, Supervision, Funding acquisition, Validation, Project administration

## Author ORCIDs

Parisa Asghari (iD) https://orcid.org/0000-0003-2285-3242
David RL Scriven (iD) https://orcid.org/0000-0003-1828-1405
Filip van Petegem (iD) https://orcid.org/0000-0003-2728-8537
Edwin DW Moore (iD) https://orcid.org/0000-0001-7519-5592

## Ethics

Animal experimentation: This study was performed in strict accordance with the recommendations provided by the Canadian Council on Animal Care. All of the animals were handled according to a protocol (A17-0040) approved by the Animal Care Committee of the University of British Columbia.

## Decision letter and Author response

Decision letter https://doi.org/10.7554/eLife.51602.sa1
Author response https://doi.org/10.7554/eLife.51602.sa2

# Additional files

## Supplementary files

- Source code 1. Program to fit RyR2 tetramers to tomographic images.
- Source code 2. Program to analyse the significance between cumultative distribution functions.
- Transparent reporting form

## Data availability

All data generated or analysed during this study are included in the manuscript and supporting files. Source data files have been provided where required.

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
