## [Decision Letter]

**Acceptance summary:**

Asghari et al. describe new results concerning the variable/changeable distribution of RyRs in response to a number of stimuli and chemical modifications. The manuscript shows that both Ca^2+^ spark properties and CRU configurations are altered in myocytes when exposed to saturating concentrations of FKBPs and/or a phoshporylation cocktail. Ca^2+^ spark properties change as well as RyR orientations within clusters. The RyR orientations are determined from EM tomograms. Additional data on peripheral cluster outlines is provided by optical super-resolution microscopy.

**Decision letter after peer review:**

Thank you for submitting your article "Cardiac ryanodine receptor distribution is dynamic and changed by auxiliary proteins and post-translational modification" for consideration by *eLife*. Your article has been reviewed by three peer reviewers, and the evaluation has been overseen by a Reviewing Editor and Kenton Swartz as the Senior Editor. The following individual involved in review of your submission has agreed to reveal their identity: Bjorn Knollmann (Reviewer #3). The reviewers have discussed the reviews with one another and the Reviewing Editor has drafted this decision to help you prepare a revised submission.

Summary:

This manuscript by Asghari et al. reports the distribution, arrangement, and cluster size of the type II ryanodine receptor (RYR2) tetramers under various conditions. This group has previously reported that phosphorylation and low concentrations of magnesium moved the tetramers into a predominantly checkerboard arrangement, whereas high concentrations of magnesium induced a dense side-by-side configuration. The present work confirms and extends their early findings and shows that FK506 binding proteins (FKBP12 and FKBP12.6) moved the tetramers into a largely side-by-side configuration and decreased the cluster size. In contrast, phosphorylation even in the presence of FKBPs moved the tetramers into a checkerboard-like configuration and increased the cluster size. By correlating the effects of FKBPs and phosphorylation with their tetramer arrangements, the authors propose that the side-by-side configuration of tetramer arrangement is associated with channel inhibition, whereas the checkerboard-like configuration is associated with channel activation.

Essential revisions:

1) The authors should show cluster rearrangements together with changes in Ca sparks in intact cells in response to strong phosphorylation. This would go a long way to demonstrate that the structural rearrangement modulate RyR function in a more physiologic setting. Importantly, these results show that hyperphosphorylation takes precedence over excess FKBP12 or 12.6." How do you reconcile this statement with some reports that phosphorylation itself leads to reduced binding of FKBPs to RyR2? Is it phosphorylation itself or is it consequent unbinding of FKBP's. How much FKBP is still bound to phospho-RyR2? I think this is shown in Supplementary Figure 6, but that figure is not referenced in the text. Is it a co-IP? This is a critical issue that needs to be addressed. Basically, does phosphorylation remove the immunophilins from the RyR2 tetramers or not? In addition to co-IP, this could/should be done with fluorescent-labelled FKBP.

2) Statistical evaluation of Ca^2+^ spark experiments and RyR distribution experiments. Recently, Sikkel et al. [1] published an analysis of Ca^2+^ spark data and included a discussion of general experiments with myocytes. They concluded that it may be required to conduct a hierarchical analysis to draw valid conclusions from myocyte experiments. The authors should show that their results hold up to such a hierarchical data analysis.

3) Determination of tetrad position and orientation in tomograms. The procedures used to determine these parameters are not clear to this reviewer. In the Materials and methods the authors say "An individual tetramer's position relative to its neighbors was classified using a custom-written C++ program, 'RyR_fit', as previously described (Asghari et al., 2014).". I consulted Asghari et al., 2014, and could not see any reference to this or a similar software in the main text or supplementary of Asghari et al., 2014. I downloaded the C++ source of RyR_fit from the supplementary of this manuscript to learn more but had difficulty to decipher answers to my questions from the lengthy source listing. In particular, is the extraction of tetrad positions and orientations fully automatic? Alternatively, does the software require the user to choose positions and orientations? A mixture of the two? In addition, with what precision/error can these parameters be determined? For example, looking at Figure 2F: for several tetrads it is not clear how the "smudges" shown can be related to a precise orientation? A more detailed description of the procedures used, particularly the role of a human operator or the robustness of an automatic procedure, should be added – including a procedure to estimate the precision with which the orientation and centre of a tetramer can be determined.

4) Actual resolution in dSTORM data. There are now a set of tools, in particular Fourier ring correlation, that can estimate the actual resolution in super-res images. This would be important to carry out to establish at what resolution level this data provides reliable information. In my experience dSTORM data often has resolution worse than 50 nm which limits conclusions that can be drawn about detailed cluster shape changes. It is important to add such estimates and then possibly re-evaluate the information that is accessible from this data.

5) "Importantly, these results show that hyperphosphorylation takes precedence over excess FKBP12 or 12.6." How do you reconcile this statement with some reports that phosphorylation itself leads to reduced binding of FKBPs to RyR2? Is it phosphorylation itself or is it consequent unbinding of FKBP's. How much FKBP is still bound to phospho-RyR2? I think this is shown in Supplementary Figure 6, but that figure is not referenced in the text. Is it a co-IP? This is a critical issue that needs to be addressed. Basically, does phosphorylation remove the immunophilins from the RyR2 tetramers or not? In addition to co-IP, this could/should be done with fluorescent-labelled FKBP.

6) The major novelty of the work is the discovery of the tremendous plasticity of the RyR2 cluster. The authors should quantify the degree of movement in more detail. What is the time course? Distances?

7) Determining the mechanism responsible for the RyR2 movements is beyond the scope of the work but the authors should discuss possible mechanisms in more detail. For example, what is the role of Bin-1 or other molecules postulated to be important for dyad formation/assembly.

---

## [Author Response]

Essential revisions:1) The authors should show cluster rearrangements together with changes in Ca sparks in intact cells in response to strong phosphorylation. This would go a long way to demonstrate that the structural rearrangement modulate RyR function in a more physiologic setting.

We are preparing a manuscript to address the hypothesis that phosphorylation of RyR2 alone, rather than other targets, is required for the functional and structural changes that we’ve observed. We used 300 nM isoproterenol as the stimulus, rather than the phosphorylation cocktail, and the data will be in that paper. The results are identical to what we have reported in this manuscript and the Asghari et al., 2014 paper.

Importantly, these results show that hyperphosphorylation takes precedence over excess FKBP12 or 12.6." How do you reconcile this statement with some reports that phosphorylation itself leads to reduced binding of FKBPs to RyR2? Is it phosphorylation itself or is it consequent unbinding of FKBP's. How much FKBP is still bound to phospho-RyR2? I think this is shown in Supplementary Figure 6, but that figure is not referenced in the text. Is it a co-IP? This is a critical issue that needs to be addressed. Basically, does phosphorylation remove the immunophilins from the RyR2 tetramers or not? In addition to co-IP, this could/should be done with fluorescent-labelled FKBP.

Multiple pieces of information indicated that RyR2 phosphorylation, or hyperphosphorylation, takes precedence. Neither protein dissociated from RyR2 to a significant extent as demonstrated by our Western Blot (Figure 7), and by our spark analyses and superresolution results. These are discussed in the subsection “Phosphorylation”. They agree with others who have performed co-IP experiments and have examined fluorescent FKBP derivatives concluding there was no significant dissociation.

2) Statistical evaluation of Ca^2+^ spark experiments and RyR distribution experiments. Recently, Sikkel et al. [1] published an analysis of Ca^2+^ spark data and included a discussion of general experiments with myocytes. They concluded that it may be required to conduct a hierarchical analysis to draw valid conclusions from myocyte experiments. The authors should show that their results hold up to such a hierarchical data analysis.

Our results do hold up and the conclusions were unchanged. We reanalysed our data using the Sikkel et al. methodology as requested, but we differed in our handling of the probability with multiple comparisons and used the sequential Holm-Bonferroni rather than the simple Bonferonni correction. We have been in contact with the R-program’s author, Dr. James Howard (the 3^rd^ author on the Sikkel et al. manuscript), and he agrees with us that the modified correction has greater statistical power. The statistical analysis is discussed in the subsection “Statistical Analyses”. There were minor changes in the results and these are reflected in Figure 1B and its source data, and in various places throughout the text.

3) Determination of tetrad position and orientation in tomograms. The procedures used to determine these parameters are not clear to this reviewer. In the Materials and methods the authors say "An individual tetramer's position relative to its neighbors was classified using a custom-written C++ program, 'RyR_fit', as previously described (Asghari et al., 2014). ". I consulted Asghari et al., 2014, and could not see any reference to this or a similar software in the main text or supplementary of Asghari et al., 2014. I downloaded the C++ source of RyR_fit from the supplementary of this manuscript to learn more but had difficulty to decipher answers to my questions from the lengthy source listing. In particular, is the extraction of tetrad positions and orientations fully automatic? Alternatively, does the software require the user to choose positions and orientations? A mixture of the two? In addition, with what precision/error can these parameters be determined? For example, looking at Figure 2F: for several tetrads it is not clear how the "smudges" shown can be related to a precise orientation? A more detailed description of the procedures used, particularly the role of a human operator or the robustness of an automatic procedure, should be added – including a procedure to estimate the precision with which the orientation and centre of a tetramer can be determined.

We have made adjustments to Figure 2 and to the text in the Results to clarify the procedure.

RyR_fit is used to manually place a square of the appropriate size and orientation over a tetramer once the operator has identified it using Amira. RyR_fit is essential for classifying the tetramers’ positions (side-by-side, checkerboard, both and isolated), for generating the NND and for determining the area occupied by the tetramers using α shapes (Figure 6D).

We estimated that the positioning error for a tetramer was ~3 nm; the basis for this estimate, and other possible errors, is discussed in detail in Asghari et al., 2014, pg 258. We have added this information to the subsection “Classifying RyR2 Positions”.

New high resolution cryo-EM structures of RyR2 show that the tetramers are 27 nm on a side (Peng et al., 2016), rather than 29 nm, which was the best estimate at the time Asghari et al. was published. It is notable that this difference has not affected our conclusions. Comparing the results from Asghari et al., 2014 to the results presented here, Control cells still show a bimodal distribution in their NND, and we see the same changes in distribution in response to agents that moved the tetramers apart (phosphorylation and low Mg^2+^) and those that brought them together (immunophilins and high Mg^2+^). These are large changes, and an error of a few nm in determining the centre of the tetramer, or of degrees in their angle, has no impact on our conclusions.

4) Actual resolution in dSTORM data. There are now a set of tools, in particular Fourier ring correlation, that can estimate the actual resolution in super-res images. This would be important to carry out to establish at what resolution level this data provides reliable information. In my experience dSTORM data often has resolution worse than 50 nm which limits conclusions that can be drawn about detailed cluster shape changes. It is important to add such estimates and then possibly re-evaluate the information that is accessible from this data.

Thank you for this suggestion. Fourier ring correlation indicated that the resolution of our images was between 16 and 25 nm. This is stated in the first paragraph of the subsection “Super-resolution immunofluorescence microscopy”, and an image of the program’s output is included in Figure 6—figure supplement 1.

5) "Importantly, these results show that hyperphosphorylation takes precedence over excess FKBP12 or 12.6." How do you reconcile this statement with some reports that phosphorylation itself leads to reduced binding of FKBPs to RyR2? Is it phosphorylation itself or is it consequent unbinding of FKBP's. How much FKBP is still bound to phospho-RyR2? I think this is shown in Supplementary Figure 6, but that figure is not referenced in the text. Is it a co-IP? This is a critical issue that needs to be addressed. Basically, does phosphorylation remove the immunophilins from the RyR2 tetramers or not? In addition to co-IP, this could/should be done with fluorescent-labelled FKBP.

Please see the answer to question 1 and the subsection “Phosphorylation”.

6) The major novelty of the work is the discovery of the tremendous plasticity of the RyR2 cluster. The authors should quantify the degree of movement in more detail. What is the time course? Distances?

Comparisons of tetramer and cluster positions before and after treatment aren’t possible because we fix the myocytes for examination by both tomography and superresolution immunofluorescence microscopy. Each of the experimental treatments involves different cells, so we cannot track individual tetramers with either technique, making it impossible to quantify the distance and time course of their movements.

Since the first time point we examined was 10 minutes after the application of the treatments we must assume that the changes occurred within this period. This is now stated in the last paragraph of the Discussion.

7) Determining the mechanism responsible for the RyR2 movements is beyond the scope of the work but the authors should discuss possible mechanisms in more detail. For example, what is the role of Bin-1 or other molecules postulated to be important for dyad formation/assembly.

A brief examination of the role of BIN1 as a possible mechanism of tetramer translocation has been added to the last paragraph of the Discussion.